# Differential activation mechanisms of lipid GPCRs by lysophosphatidic acid and sphingosine 1-phosphate

Shian Liu[1,5], Navid Paknejad [2,5], Lan Zhu[3], Yasuyuki Kihara[4], Manisha Ray[4], Jerold Chun [4], Wei Liu[3], Richard K. Hite [2✉] & Xin-Yun Huang [1✉]

Lysophospholipids are bioactive lipids and can signal through G-protein-coupled receptors (GPCRs). The best studied lysophospholipids are lysophosphatidic acid (LPA) and sphingosine 1-phosphate (S1P). The mechanisms of lysophospholipid recognition by an active GPCR, and the activations of lysophospholipid GPCR–G-protein complexes remain unclear. Here we report single-particle cryo-EM structures of human S1P receptor 1 (S1P$_1$) and heterotrimeric G$_i$ complexes formed with bound S1P or the multiple sclerosis (MS) treatment drug Siponimod, as well as human LPA receptor 1 (LPA$_1$) and G$_i$ complexes in the presence of LPA. Our structural and functional data provide insights into how LPA and S1P adopt different conformations to interact with their cognate GPCRs, the selectivity of the homologous lipid GPCRs for S1P versus LPA, and the different activation mechanisms of these GPCRs by LPA and S1P. Our studies also reveal specific optimization strategies to improve the MS-treating S1P$_1$-targeting drugs.

[1] Department of Physiology and Biophysics, Weill Cornell Medical College of Cornell University, New York, NY 10065, USA. [2] Structural Biology Program, Memorial Sloan Kettering Cancer Center, New York, NY 10065, USA. [3] School of Molecular Sciences and Biodesign Center for Applied Structural Discovery, Arizona State University, Tempe, AZ 85287, USA. [4] Sanford Burnham Prebys Medical Discovery Institute, 10901 N Torrey Pines Rd, La Jolla, CA 92037, USA. [5]These authors contributed equally: Shian Liu, Navid Paknejad. ✉email: hiter@mskcc.org; xyhuang@med.cornell.edu

Lysophospholipids are bioactive lipids with a phosphate head group and a single hydrophobic fatty acyl chain[1–3]. These lipids can function as endogenous extracellular ligands for G-protein-coupled receptors (GPCRs), thereby triggering intracellular signaling pathways[2,4–6]. The best studied lysophospholipids are lysophosphatidic acid (LPA) and sphingosine 1-phosphate (S1P)[7]. LPA receptor 1 (LPA₁) was the first receptor identified for any lysophospholipids, and is one of six recognized LPA receptors[8–10]. Studies of mice lacking the LPA₁ gene *Lpar1* have revealed physiological and pathophysiological functions of LPA₁ including neural development and function, bone homeostasis, pain, hydrocephalus autoimmune disorders, and development and progression of fibrosis[1,2,11–14]. LPA₁ is a therapeutic target for treating idiopathic pulmonary fibrosis that can impact COVID-19 patients, while being further assessed in systemic sclerosis and related fibrotic diseases, as well as obesity and stress incontinence[15–19]. The structure of an inactive state of LPA₁ was solved by X-ray crystallography through co-crystallization of an engineered LPA₁ and synthetic non-lipid antagonists[20]. However the mechanisms of LPA recognition by an active LPA receptor, and the activation of an LPA receptor–G-protein complex by the lipid agonist remain unclear.

Multiple sclerosis (MS) is a devastating disease of the central nervous system characterized by the progressive destruction of the myelin sheath of the axons by the immune cells, leading to neuronal degeneration[21]. The cause remains elusive, and treatments are few[21]. Among the drugs used to treat MS is the immunosuppressors targeting S1P receptors[22,23]. There are five S1P receptors[24]. The MS drug efficacy is through S1P₁. S1P₁ gene knock-out mice have been instrumental for determining the importance of S1P₁ for immune cell trafficking[25]. When S1P₁ was deleted specifically in T cells, maturing thymocytes were unable to emigrate out of the thymus into the circulation, leading to a systemic lack of T cells[25]. When in the periphery, T cells with lower-than-normal levels of S1P₁ were retained in spleen and lymph nodes, and were deficient in lymph and blood[25]. Immature B cells also required S1P₁ for their exit out of the bone marrow, where they are generated, and for entry into the circulation. Their egress from bone marrow parenchyma into sinusoids was severely impaired in B cell-specific S1P₁-null mice[26]. Circulating S1P₁-deficient B cells were trapped in lymph nodes in fetal liver chimeric mice[26]. These studies established that S1P₁ is necessary for lymphocytes to exit primary and secondary lymphoid organs. S1P₁ couples to and signals through Gi[27]. An X-ray crystal structure of the inactive state S1P₁ bound with the antagonist ML056 had been reported[28].

Here we report single-particle cryo-EM structures of the signaling complexes of human S1P₁ and Gi in the presence of S1P or Siponimod (a MS treatment drug), as well as human LPA₁ and G$_i$ complexes formed with bound LPA. The structures provide insights into how the structurally similar lipids S1P and LPA adopt different conformations to activate their cognate GPCRs, how the GPCRs discriminate the two similar lysophospholipids, how the receptors accommodate lysophospholipid agonists through reorganization of inter-helical contacts, and how different lipid agonists activate their receptors (a unique mechanism for lipid GPCR activation that differs from other class A GPCRs). Furthermore, multivariate 3D analysis further reveals the dynamic nature of the interaction between the receptors and the G-proteins, and distinct conformational states of the complexes. Moreover, recent developments of MS therapeutic drugs targeting S1P₁ showed various specificity, while the exact structural basis and further optimization strategy are not certain. Our study reveals the similarities and differences of the MS-targeting drugs interacting with S1P receptors, and provides specific routes to improve drug specificity towards S1P receptors.

## Results

**Overall structures of agonist bound S1P₁-G$_i$ and LPA₁-G$_i$ complexes.** To understand how lysophospholipid agonists interact with their cognate GPCRs, and how they activate the GPCR–G-protein signaling complexes, we solved the single-particle cryo-EM structures of the human S1P₁-G$_i$ (Gα$_{i1}$Gβ₁Gγ₂) complex in the presence of d18:1 S1P and Siponimod, as well as the human LPA₁-G$_i$ complex in the presence of 18:1 LPA. The d18:1 S1P and 18:1 LPA are the most abundant endogenous lipid agonists for S1P₁ and LPA₁, respectively, while Siponimod represents a synthetic compound specifically targeting S1P₁ and S1P₅ among the five S1P receptor subtypes, and has recently been approved to treat MS. Three structures were determined to the resolution at 2.8 Å, 3.0 Å and 2.8 Å, respectively, enabling us to unambiguously place their agonists and the majority of side chains to identify rotameric changes (Supplementary Figs. 1–5, and Supplementary Tables 1 and 2). The overall folds of S1P₁ and LPA₁ are similar with a short N-terminal helix capping the orthosteric binding site from the extracellular side (Fig. 1; Supplementary Figs. 4 and 5). A unique glycosylation modification on the cap segment of S1P₁ was previously shown to be critical for the correct protein trafficking, but does not appear to participate in the receptor activation in our structures (Fig. 1d and e; Supplementary Fig. 4).

**Different lipid binding modes in S1P–S1P₁ and LPA–LPA₁.** The orthosteric binding sites between the S1P₁ and LPA₁ receptors show distinct architectures. In S1P₁, the ligand-binding site is a cylindrical concavity, 30° tilted relative to the membrane, pointing deep into the helical bundle core until the TM4-TM5 cleft, where the length of the pocket almost spans the entire outer leaflet of the lipid bilayer (Fig. 2a and c). On the contrary, the ligand-binding site in LPA₁ shows a pouch-like side pocket (Fig. 3a and c). The phosphate head groups of both types of lysophospholipids face extracellularly and interact with the cap, TM2, TM3 and TM7 (Figs. 2b and 3b). The hydrophobic tail of S1P extends to the end of the orthosteric binding pocket surrounded by TM3-7 (Fig. 2b), while in LPA₁ TM4 is not involved in coordinating LPA (Fig. 3b). Unexpectedly, the side binding pockets of S1P₁ fold into a trefoil shape (marked as Sites B1, B2 and B3 in Fig. 2c and d, and Supplementary Fig. 6) where the tail of S1P and the cyclohexyl group of Siponimod point into B2 (Fig. 2c and d). The trifluoromethyl group of Siponimod is specifically located at B1, but the smallest B3 site is left unoccupied (Fig. 2d). It was previously shown that incorporating the cyclohexyl group rendered Siponimod the increased specificity for S1P₁[29]. The B3 site discovered in our structures can be further exploited to design drugs to achieve optimized receptor subtype specificity and reduced side effects.

Due to the large size of the orthosteric binding sites in the two receptors, a significant number of residues are involved in the binding of these lysophospholipid ligands. In S1P₁, the hydrophobic side pocket is surrounded by residues including L195 (from the extracellular loop 2 (ECL2)), M124, F125, L128, S129, V132 and F133 (in TM3), L174 (in TM4), C206, V209, F210 and L213 (in TM5), W269 and L272 (in TM6), as well as L297 (in TM7) (Fig. 2e and f). The phosphate head group is coordinated by Y29 and K34 from the cap, R120 (in TM3), T109 and the backbone nitrogen of G106 (in ECL1) (Fig. 2g). A previously functionally identified key residue, E121 (in TM3), does not form any direct contacts with S1P in our structure; instead, their interaction is mediated by N101 (in TM2) in the coordination of the amino group of S1P (Fig. 2g). This interaction is joined by R120 to form a hydrogen bond triad in the coordination of Siponimod (Fig. 2h).

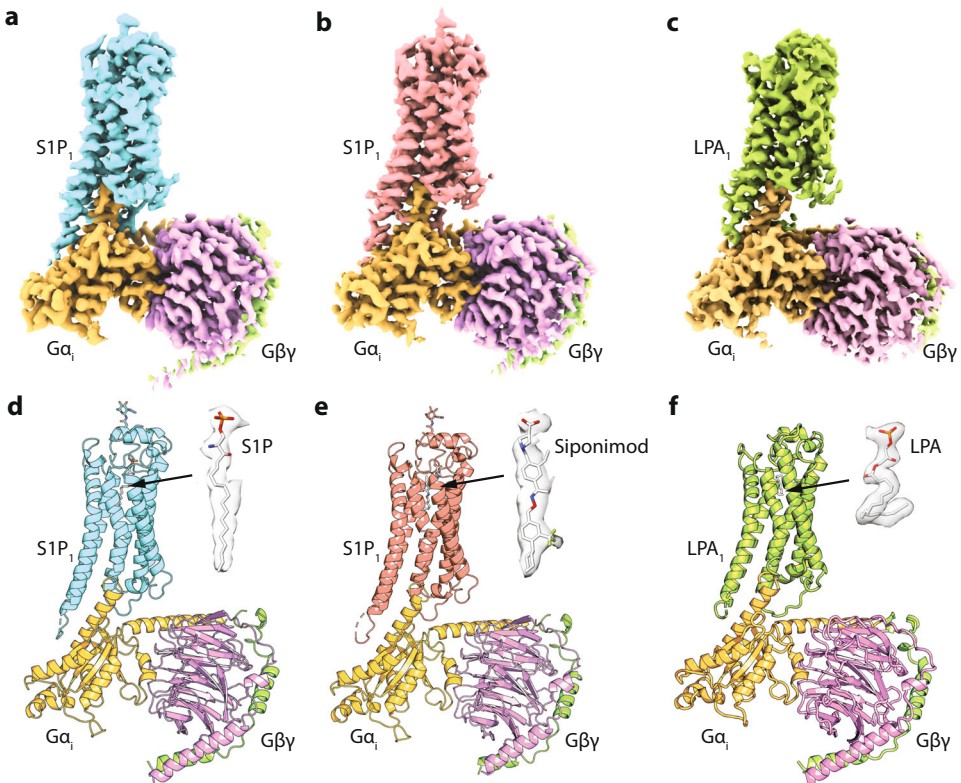

**Fig. 1 Cryo-EM structures of the S1P–S1P₁–Gᵢ complex, the Siponimod–S1P₁–Gᵢ complex, and the LPA–LPA₁–Gᵢ complex. a–c** The three-dimensional density maps of the complexes. **d–f** Cartoon presentations of the three complex structures with S1P, Siponimod, or LPA shown in spheres inside the orthosteric pockets, as well as their density maps on the sides. S1P₁ is colored in cyan or salmon, LPA₁ in green, Gαᵢ in orange, Gβ in purple, and Gγ in light green.

In LPA₁, the long unsaturated fatty acyl chain of LPA adopts a U-shaped conformation extending into the receptor and then bending backwards (Fig. 3d–f). Residues I128, D129 and L132 (in TM3), as well as Y206, L207 and Y202 (in TM5) contact the descending half of the fatty acyl chain (Fig. 3d). At the bottom of the cavity, W210 (in TM5) pushes against the bending region of LPA (Fig. 3d and e). On the other side of the pocket, W271, G274, L277 and L278 (in TM6), L297 and A300 (in TM7), as well as M198 from ECL2 interact with the terminal ascending portion of the fatty acyl chain (Fig. 2e). The small pocket encaging the terminus of LPA is largely attributed to the absence of side chain from G274 at TM6, whereas in S1P₁ it is occupied by L272 at the equivalent position (Figs. 2f and 3e). Residues, Y34 and K39 (from cap), T109 and T113 (from ECL1), R124 (in TM3), and K294 (in TM7) interact with the negatively charged phosphate head group of LPA (Fig. 3f). Residue Q125 was proposed previously to directly bind to the phosphate head group[20], but in our structure, it forms a hydrogen bond with the acyl carbonyl oxygen of LPA (Fig. 3f). To validate the physiological functions of the structurally identified residues, we mutated several lipid-interacting residues and performed functional studies. Both S1P₁ and LPA₁ can activate Gi to decrease the cellular levels of cAMP, and these mutations impaired the agonist-induced cAMP signaling (Supplementary Fig. 7). Several key mutations of LPA₁ also confirmed to hinder the LPA association measured by the ligand-binding assays (Supplementary Fig. 7).

Unlike other types of GPCR ligands, lysophospholipids have long flexible acyl chains (Supplementary Fig. 8). From our structural studies, several factors contribute to the selectivity of these receptors for S1P or LPA. First, the shape of the orthosteric binding site determines the types of lysophospholipid. Both S1P and LPA have been co-crystallized with other proteins, where their fatty acid tails were found to adopt curled conformations (Supplementary Fig. 8). LPA exhibits more flexibility (Supplementary Fig. 8b). For example, LPA bound to LPA₁ shows a marked rotation in respect to the ester bond compared to its conformation complexed with autotaxin, an LPA producing enzyme (Supplementary Fig. 8b). Together with the unsaturated bond in the middle of the acyl chain, LPA can adapt to the pouch-shaped pocket of LPA₁ (Fig. 3c). Second, the size of the orthosteric binding site discriminates against the length of the lipids. For example, S1Ps with the acyl chains shorter than 16 or 18 carbons and short lipid mimicking compounds, such as ML056, failed to activate S1P₁[30]. Similarly, LPAs with shorter acyl chains such as myristoyl (14:0), lauroyl (12:0), capric (10:0), or caproic (6:0) are less potent agonists of LPA₁[31].

**Different activation mechanisms between S1P₁ and LPA₁.** To understand the activation of GPCRs by lysophospholipids, we compared the structures of active to inactive S1P₁, and active to inactive LPA₁ (Figs. 4 and 5). The two receptors manifest different changes on their ECLs upon the agonist association (Fig. 4). In S1P₁, the ECL0 is a flexible loop flipped to pack against ECL2 in the active conformation, which leads to an opening of the ligand access vestibule (Fig. 4a and b). The same region of LPA₁, however, is kept open between the inactive and active states (Fig. 4c). Instead, the main change in LPA₁ is manifested by ECL3 approaching closer to the N-terminal cap segment that causes L290 to flip upward from a buried position in the inactive state and to make a direct interaction with the backbone of the cap segment (Fig. 4c). Additionally, H40 was previously proposed to be involved in the phosphate coordination based on the inactive LPA₁ structure, whereas here it is displaced

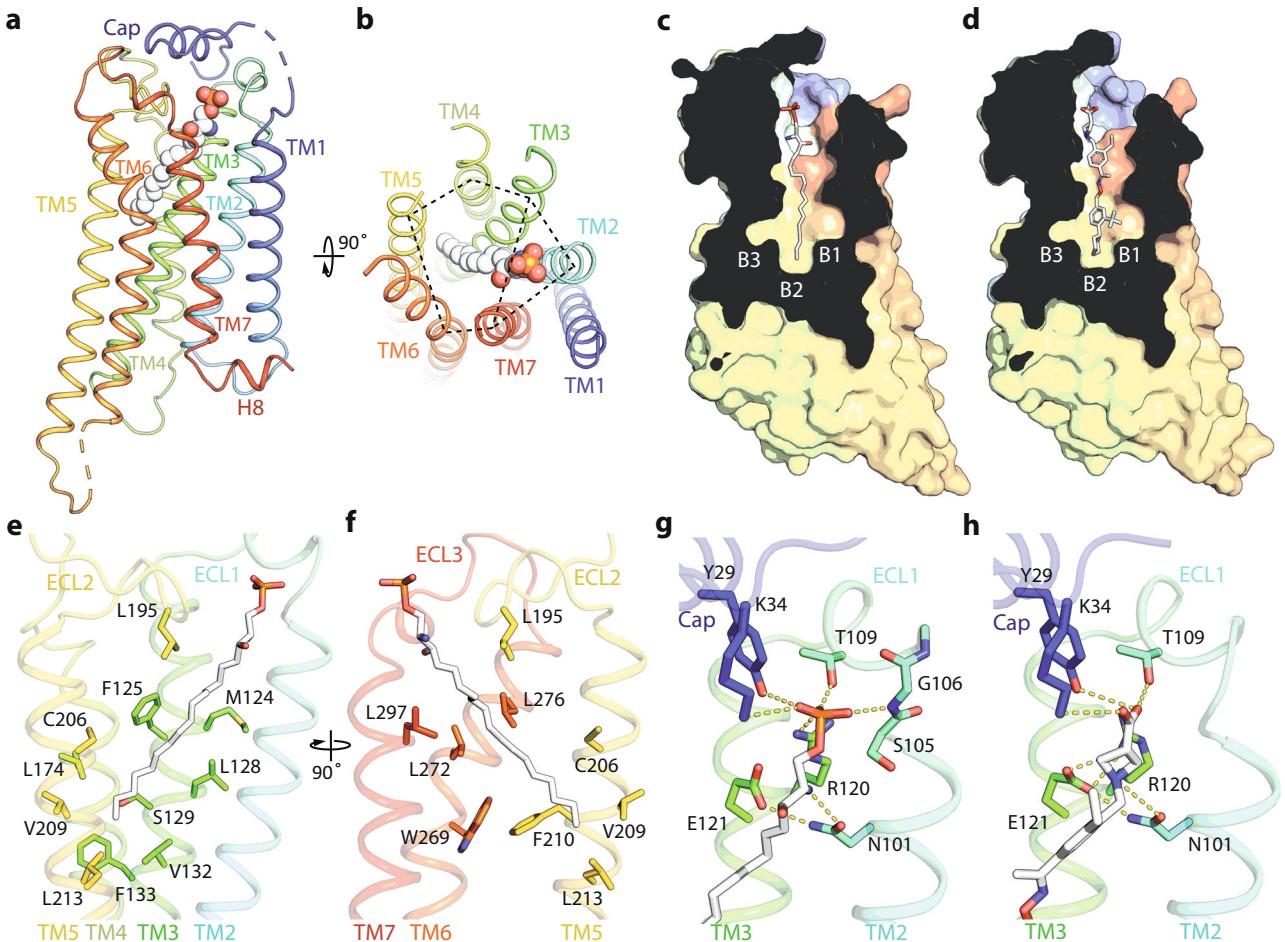

**Fig. 2 Interactions between S1P or Siponimod and S1P₁. a, b** Cartoon representation of S1P₁ with the bound S1P in spheres, and rotated 90° to show the two binding modules of transmembrane (TM) helices indicated by dashed lines. **c, d** Slice through views of the ligand-binding pockets of S1P₁ with the side binding pockets B1, B2 and B3 indicated. **e, h** Residues involved in interacting with the hydrocarbon chain (**e–h**) or the polar head group (**g, h**) of S1P or Siponimod are shown in sticks.

from the ligand-binding pocket by the shift of K294 (Fig. 4c). These distinct features indicate that ligands may enter S1P₁ and LPA₁ differently.

The movements of the extracellular parts of the two receptors propagate through the orthosteric binding sites. Remarkably, compared with the lipid-mimicking antagonist ML056, the longer agonist S1P or Siponimod extends the binding pocket of S1P₁ deeper into the membrane in a trefoil shape (Fig. 5a, and Supplementary Fig. 8). From the inactive to the active conformation of S1P₁, one essential change is the flipping of the side chain of F210 from the TM3-TM5 interface to the TM5-TM6 interface (Fig. 5c–e, Supplementary Fig. 9). Along with that, W269 and F273 rotate their side chains leading to an expanded B1 side binding pocket in the active S1P₁, which nicely explains the increased specificity of Siponimod owing to the addition of a trifluoromethyl group (Fig. 5d and e). In coordination with F210, residues S129, F133 and V209 shift towards the agonist and enclose the end of the side pocket (Fig. 5d and e). Finally, the emerging of the B3 side binding pocket is due to the rotameric change of L128, which packs against L297 from TM7 in the inactive state (Fig. 5c).

In contrast, the overall shape of the side binding pocket of LPA₁ has not undergone as much changes as in S1P₁ during its transition from the inactive state to the active state (Fig. 5b, f and g, Supplementary Fig. 9). This is largely due to the bulky side chain of W210 preventing LPA from reaching deeper into

the membrane across the TM3-TM5 interface (Fig. 5g). Instead, the rotameric changes of L132 and W210 accommodate the sharp turn of the fatty acyl chain, and the conserved movement of W271, which denotes the opening of TM6 in class A GPCRs[32]. These allow the position of the acyl terminus of LPA between the TM6-TM7 interface (Fig. 5g). As TM7 shifts into the core of LPA₁, Y102 and D301 swap their side chain positions to support the bottom of LPA (Fig. 5g). The changes around the orthosteric binding site propagate to TM5 (Fig. 5h and i). In S1P₁, the flip of F210 causes the rotameric change of L214, accompanied by the translational movement of W269 and F265 in TM6 to form new contacts (Fig. 5h). A series of new contacts are formed at four equivalent positions of LPA₁ below residue W210 in the active conformation (Fig. 5i). To interrogate the functional roles of these new contacts, we created F210A and F273A mutants in S1P₁ and performed cAMP assays in response to S1P (Fig. 5j) and Siponimod (Fig. 5k). Both mutants impaired the receptor signaling, confirming that this region is essential for the activation of the receptor. Together, the above data show that the two receptors are activated differently by S1P or LPA.

Fingolimod (FTY720) was the first oral treatment for multiple sclerosis (MS), and its effective form, phosphorylated Fingolimod, shares a similar chemical scaffold as S1P and lacks receptor specificity (Supplementary Fig. 8). Adverse effects, especially the bradycardia caused by S1P₃ activation, led to its restricted

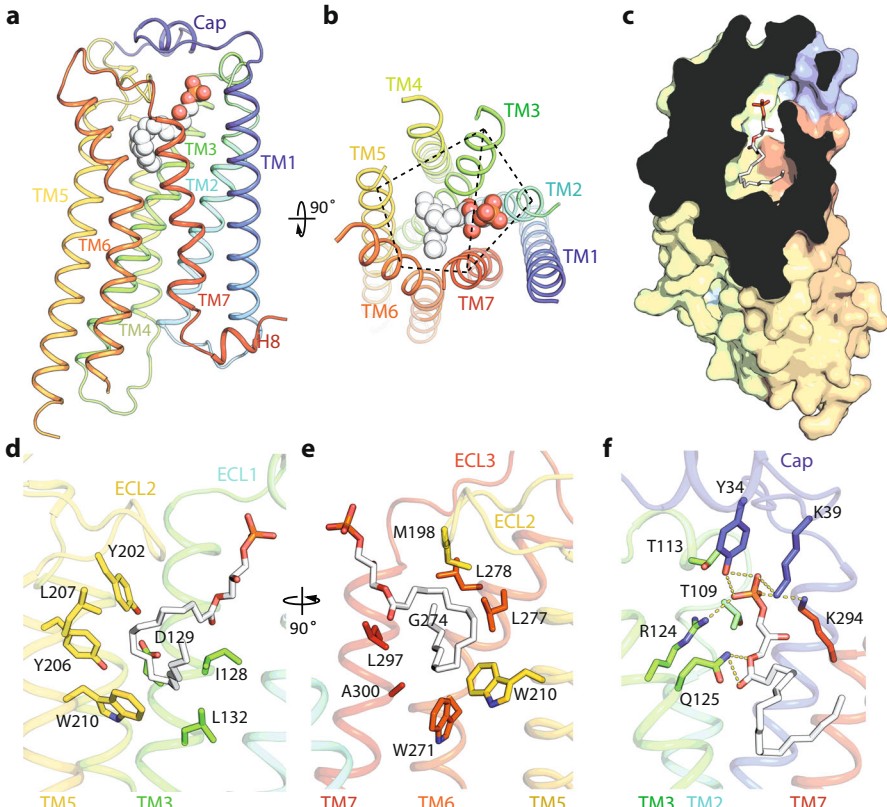

**Fig. 3 The orthosteric LPA-binding pocket of the active LPA₁. a, b** Cartoon representation of LPA₁ with the bound LPA in spheres, and rotated 90° to show the two binding modules of transmembrane (TM) helices indicated by dashed lines. **c** Slice through view of the LPA-binding pocket. **d–f** Residues involved in interacting with the hydrocarbon chain (**d, e**) or the polar head group (**f**) of LPA are shown in sticks.

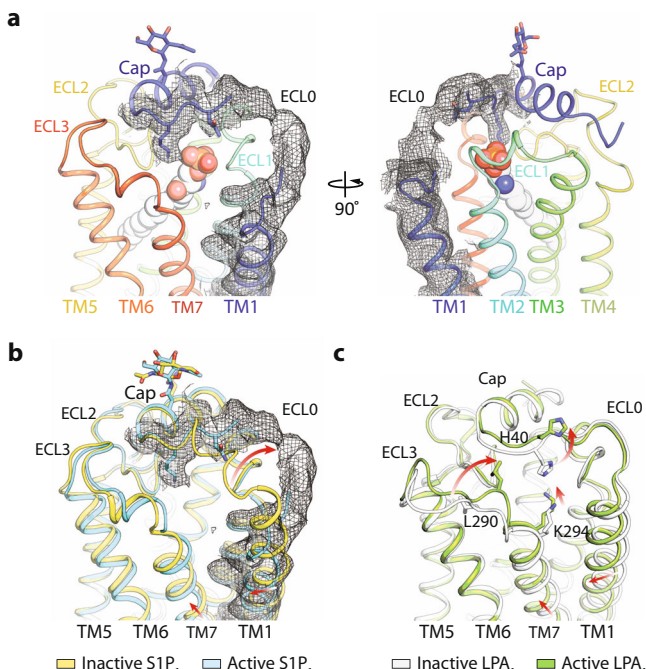

**Fig. 4 The ECL movement during the activation. a, b** The density overlaid with the model of active S1P₁ compared to inactive S1P1. **c** The model of active LPA₁ compared to inactive LPA₁. The movement is indicated by red arrows, and key residues are shown in sticks.

clinical uses. The development of the next generation of more selective S1P receptor modulators has led to the approval of Siponimod, Ozanimod and Ponesimod for the treatment of relapsing MS in 2019, 2020 and 2021, respectively (Supplementary Fig. 8). All three drugs selectively activate S1P₁ and S1P₅, over S1P₂, S1P₃ and S1P₄; however, their structural basis is unclear. Given the high amino acid sequence similarity among them, especially at the TM regions, we simulated the structures of S1P₂, S1P₃, S1P₄ and S1P₅ (Fig. 6). Docking Siponimod into the four models of S1P₂, S1P₃, S1P₄ and S1P₅ reveals major clashes from the unique phenoalanine residues in S1P₂ and S1P₃ and a minor clash from the methionine residue in S1P₄ (Fig. 6). We then generated mutations L297F, L276F, and A293M of S1P₁, to mimic the pockets in S1P₂, S1P₃ and S1P₄ (Fig. 5l). All these mutations impaired the function of S1P₁ in response to Siponimod, and thus confirming that these residues are indeed contributing to the receptor-binding specificity. The same mechanism is likely to apply to Ozanimod and Ponesimod (Fig. 6, Supplementary Fig. 8). Therefore, we hypothesize that the shape of the orthosteric binding site in the active state conformation of S1P₁ is primarily responsible for discriminating different agonists. Optimizing the chemical structures of the drugs to fully explore all B1, B2 and B3 side binding pockets will further improve the specificity to desired receptor subtypes.

**Activation of Gi by S1P₁ and LPA₁.** Having established how agonists differentially activate S1P₁ and LPA₁, we next examined how they drive the activation of Gi. Comparing to other Class A GPCR complex structures, Gi in our structures rotates closer to

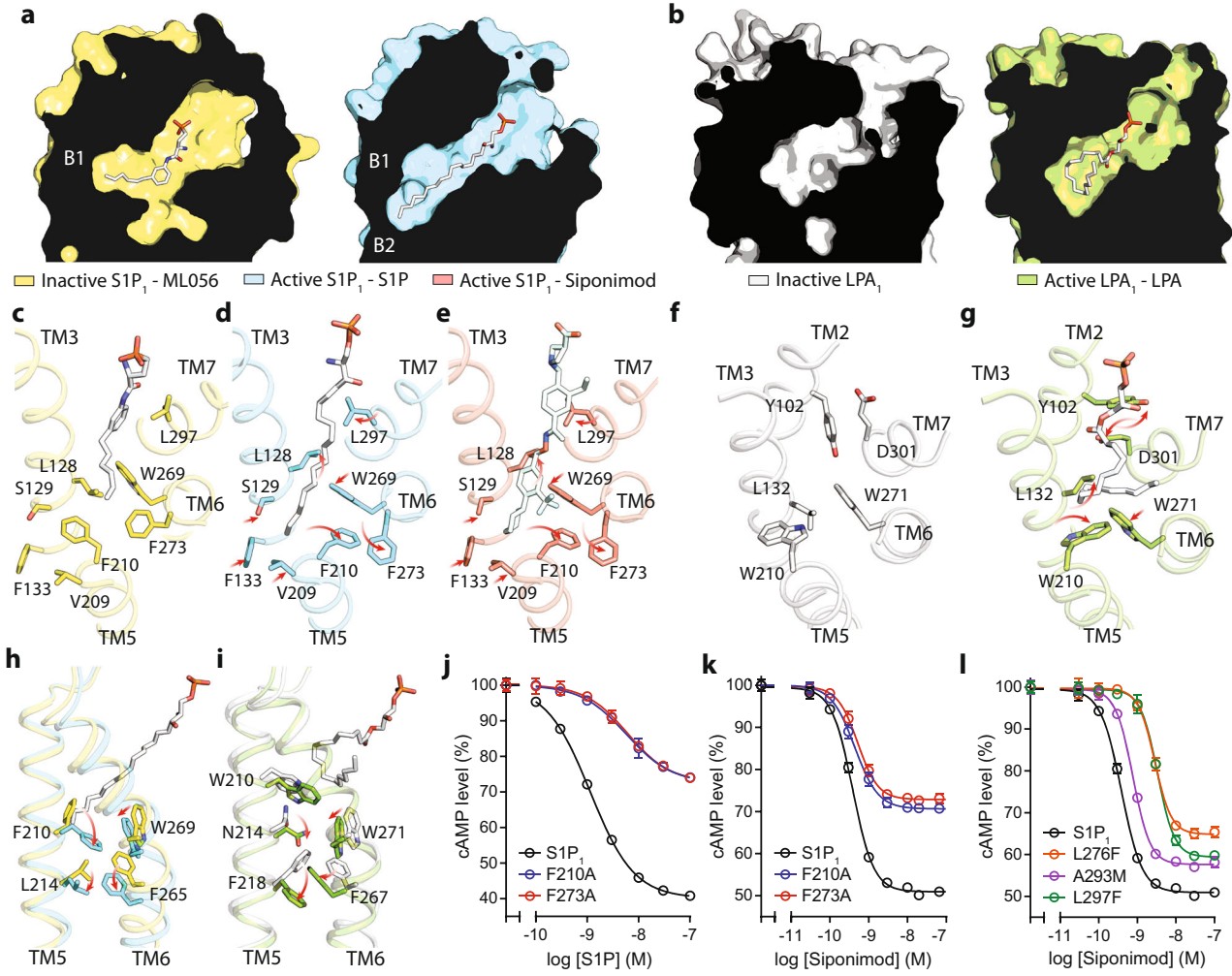

**Fig. 5 Comparison of the activation mechanisms of lysophospholipid receptors. a** Slice through views of the orthosteric ligand binding pockets of S1P₁ when bound with the antagonist ML056 (in yellow), or S1P (in cyan). **b** Slice through views of the orthosteric ligand binding pockets of LPA₁ in the inactive (in white) or active (in green) states. **c**–**e** key residues for S1P₁ activation are shown in sticks to illustrate the rotameric reorganization of their side chains at the bottom of the ligand-binding pocket, when bound with ML056 (**c**), S1P (**d**), or Siponimod (**e**). **f, g** Key residues for LPA₁ activation are shown in sticks to illustrate the rotameric reorganization of their side chains, when in the inactive unliganded state (**f**), or in the active LPA-bound state (**g**). **h, i** The conformational changes are propagated from the bottom of the ligand-binding pocket to the G-protein binding pocket near TM5 and TM6, in S1P₁ (**h**) and LPA₁ (**i**). **j**–**l** Dose–response data from cells expressing different S1P₁ constructs after stimulation with S1P or Siponimod. Data are shown as mean ± SEM of three experiments. The analysis was done using the log(agonist) vs. response function of Prism 8 (GraphPad). Source data are provided as a Source Data file.

and engages more interactions with ICL3, whereas ICL2 is an unstructured loop rather than adopting a short helical structure (Supplementary Fig. 10). To stabilize the position of Gi, S1P₁ engages several unique interactions (Fig. 7). In its ICL2, L151 is inserted into the hydrophobic pocket at the end of the N-terminal helix of Gαᵢ, and N153 forms hydrogen bonds with D350 and N347 from the α5 helix of Gαᵢ (Fig. 7a and b). In ICL3, L235 and F237 contact the hydrophobic surface of Gαᵢ, and K250 forms a salt bridge with D341 from the α5 helix of Gαᵢ (Fig. 7a and b). Furthermore, R231 located at TM5 stretches to the amino-end of TM3 helix and interacts with the carbonyl backbone through hydrogen bonding (Fig. 7b, inset).

However, we noticed that several of these interactions are absent in the LPA–LPA₁-Gi complex, which appears to be far more dynamic. 3D variability analysis (3DVA) has been recently made available to study local motion of cryo-EM maps and successfully applied to several GPCR–G-protein complexes[33]. By applying 3DVA, we resolved the rocking, twisting, and flexing motion of LPA₁ about the G-protein (Supplementary Movies 1–6).

To reveal Gi's movement in more details, 1.6 million particles were clustered in the 3DVA latent space to identify two extreme conformations. Given the high quality of the maps, we were able to build atomic models at 3.2 Å and 3.2 Å, respectively, and identify unprecedented changes of GPCR–G-protein complex (Supplementary Figs. 5 and 11). The major distinction between the two states is the relative rotation of Gαᵢ about LPA₁ in the plane of the membrane, ~5° in both directions away from the consensus structure (Fig. 8a). While the overall conformation of the complex remained the same (Supplementary Figs. 5 and 11), state a and state a' show marked differences at the interface between LPA₁ and Gi. In state a, L155 from ICL2 is inserted into the hydrophobic pocket of αN helix (Fig. 8b). Surprisingly, in state a', the side chain of L155 is turned to occupy a hydrophobic pocket underneath the α5-helix of Gαᵢ, leading to a rotameric change of N346 (Fig. 8c). Accompanying the differences of ICL2 are alterations in the conformation of F354, the extreme C-terminal residue in Gαᵢ (Fig. 8d and e). In state a, the side chain of F354 lays along the interior surface of TM6 (Fig. 8d);

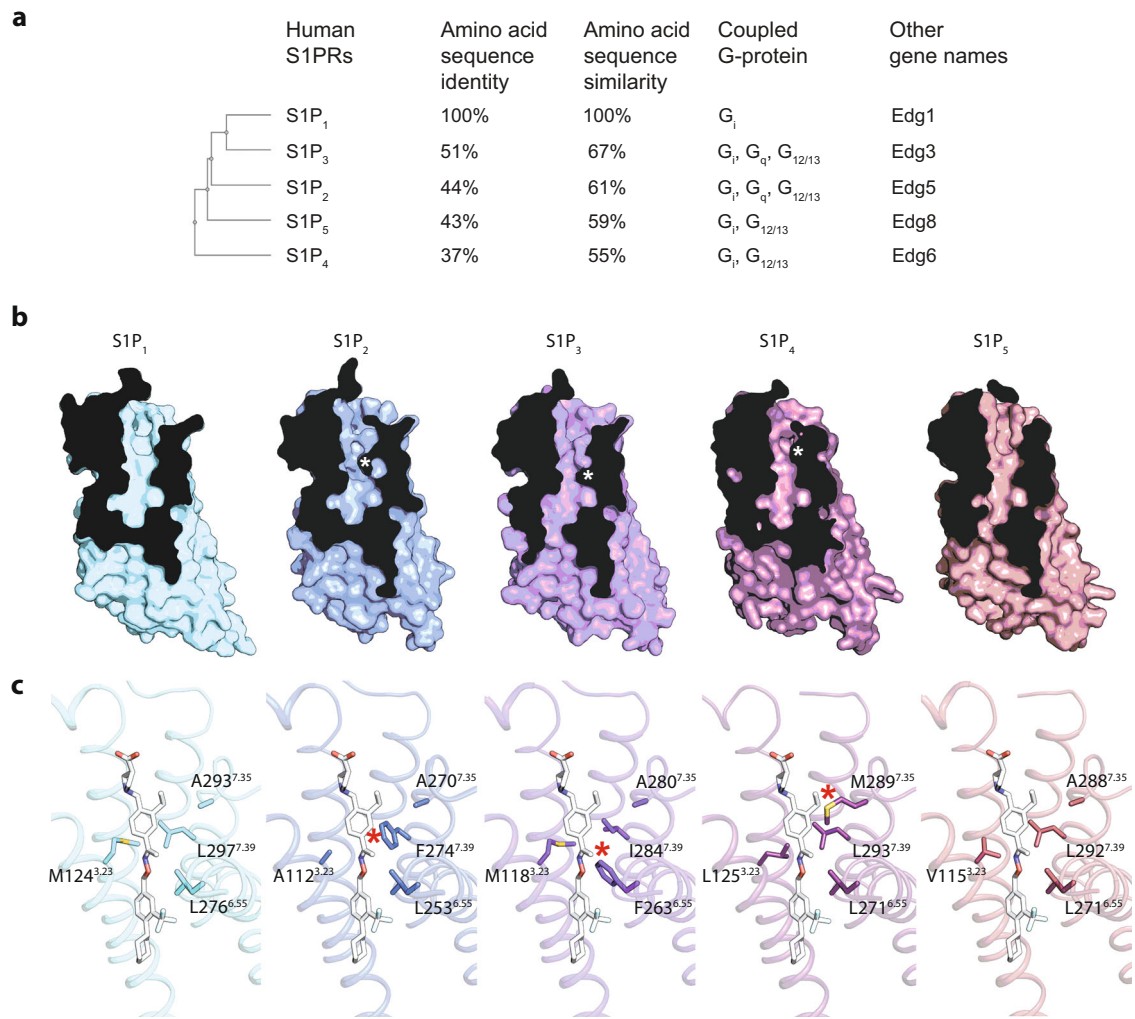

**Fig. 6 S1P receptor classification and modeled structures. a** Classification, sequence homologies, coupled G-proteins and other names of S1P GPCRs. **b** Slice views of the orthosteric binding sites of S1P₁ and the modeled S1P₂₋₅. **c** Docking of Siponimod into the five S1P receptors. * indicates the clashes.

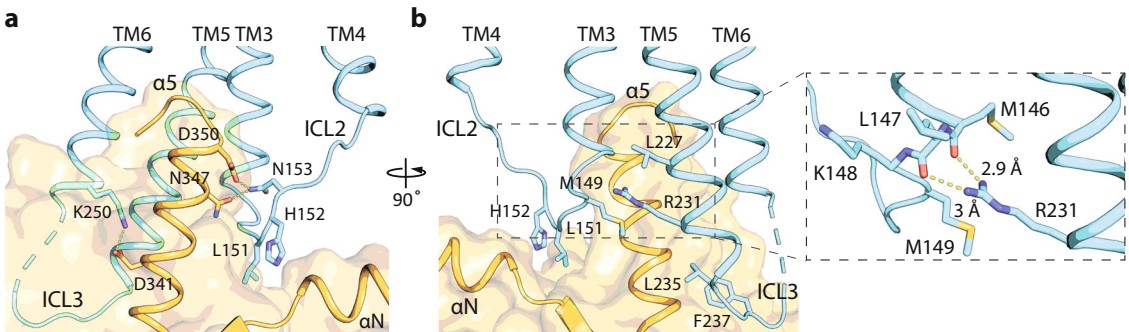

**Fig. 7 Interactions between ICL2 of S1P₁ and Gαᵢ. a** L151 in ICL2 is inserted into the hydrophobic pocket at the end of the N-terminal helix of Gαᵢ, and N153 forms hydrogen bonds with D350 and N347 from the α5 helix of Gαᵢ. **b** L235 and F237 in ICL3 contact the hydrophobic surface of Gαᵢ, and K250 forms a salt bridge with D341 from the α5 helix of Gαᵢ. R321 located at TM5 stretches to the amino-end of TM3 helix and interacts with the carbonyl backbone through hydrogen bonding.

however, in state a', F354 flips its side chain to interact with the ICL4 of LPA₁ (Fig. 8e). Since ICL2 of the receptor and the C-terminal tail of the G-protein are two critical regions involved in the activation of G-proteins, the correlated conformational changes in the LPA-LPA₁-Gi complex described here provide a mechanism of their dynamics. We also performed 3DVA on the complexes of S1P–S1P₁–Gi (Supplementary Movies 7–12) and

Siponimod–S1P₁–Gi (Supplementary Movies 13–18). Both the receptor and Gi-protein displayed flexibility in these complexes.

Lastly, we sought to understand the activation mechanism of Gi. For the Gαᵢ₁ subunit, the last three residues of α5-helix in the inactive Gi were disordered and unresolved, while they form a helix extension and interact extensively with S1P₁ and LPA₁ (Figs. 7 and 8). Moreover, the interacting network between the

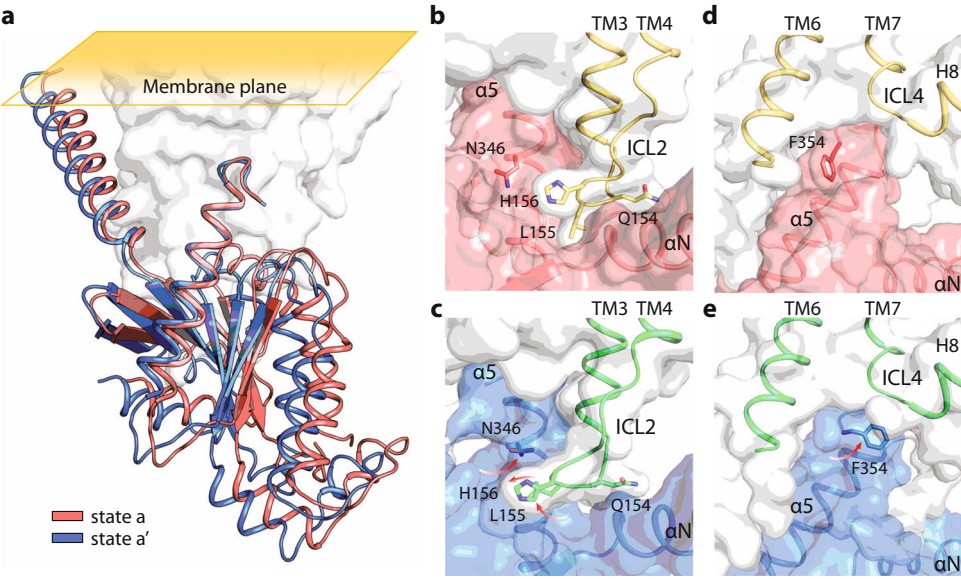

**Fig. 8 The dynamic interactions between LPA₁ and Gαᵢ. a** The superposition of State a and a' in the first component of 3DVA with the receptor shown in white surface and Gβγ removed for clarity. **b, c** Key residues for the interactions among ICL2, TM4 and TM5 of the receptor and the α5 helix of Gαᵢ are shown in sticks for comparisons overlaid with the surface view of Gαᵢ. **d, e** Comparison of the side chain rotamers of F354 at the two states.

N-terminal and C-terminal parts of Gαᵢ₁ is broken, such as the ionic interaction between D341 and K192 (Supplementary Fig. 12a). In its new position, D341 forms another ionic lock with K250 in S1P₁ (Supplementary Fig. 12a). Furthermore, in the inactive Gi heterotrimer, Q52 locks in the TCAT motif via a hydrogen bond with the backbone carbonyl of A326 and the side chain of T329 (Supplementary Fig. 12b). However, in GPCR-Gi complexes, this network is broken leading to the movements of the α1-helix, the P-loop and the TCAT motif (Supplementary Fig. 12b). As these regions form the binding pocket for GDP, disruption of these interactions leads to GDP release and Gi activation.

### Discussion

In this work, we solved, compared, and contrasted the structures of two representative lysophospholipid GPCRs that provided unprecedented information for their activation mechanisms by endogenous lipid agonists. Both LPA and S1P are amphipathic lipids with a polar zwitterionic head-group, a tri-carbon middle moiety and a hydrophobic alkyl chain. Their phosphate head-group interacts specifically with conserved structural arrangements on the extracellular side of the receptors. Positively charged residues K34 and R120 of S1P₁ as well as corresponding residues K39 and R124 of LPA₁ form salt bridges with the negatively charged phosphate, and polar uncharged residues form a secondary coordination shell also irreplaceably supporting the binding of phosphate group. Such arrangement of coordination is absent in other GPCRs such as cannabinoid receptors[34], which nicely explains why various fatty acids without a phosphate head group are unable to activate lysophospholipid GPCRs. Previously, Q125 of LPA₁ was proposed to be a phosphate interacting residue; mutating Q125, or the corresponding E121 of S1P₁, was shown to compromise the selectivity of lysophospholipids between LPA₁ and S1P₁[35]. In our structures of the active states of LPA₁ and S1P₁, however, these residues were found to recognize the carbonyl oxygen of the LPA ester bond or the amino group of S1P, suggesting the middle moiety of lysophospholipids contribute to their specificity to these receptors (Figs. 2g, h and 3f). Remarkably, the hydrophobic side pockets of two receptors reveal distinct architectures, a pouch shaped pocket in LPA₁ versus a straight slender pocket in S1P₁ (Figs. 2c, d and 3c). Owing to the bulkiness of W210 in LPA₁, its side pocket can only accept lipids that bend their fatty acid tail backwards such as the case of LPA (Figs. 3d, e, 5g and i). Unlike having a tryptophan residue in LPA₁₋₃ or CB₁₋₂, the equivalent position in S1P₁₋₅ receptors is occupied by a phenylalanine residue. A recent X-ray crystal structure of S1P₃ bound with S1P (without a G-protein) showed a similar S1P-binding mode (Supplementary Fig. 13a)[36]. Also, two recent publications reported the cryo-EM structures of the complexes of S1P₁–Gᵢ, S1P₅–Gᵢ and S1P₃–Gi (Supplementary Fig. 13b)[37,38]. Upon activation, F210 in S1P₁ rotates away from the side pocket to create extended trefoil space for long straight ligand to be accommodated (Fig. 5d and e). Hence, lysophospholipid receptors evolve into unique and intricate mechanisms to specifically recognize all three components of a lysophospholipid in order to achieve their selective activation. In addition, W210 in LPA₁ or F210 in S1P₁ is in a central position to propagate the agonist binding to the movement of TM6 (thus the activation of the receptor) (Fig. 5h and I, Supplementary Fig. 9), where four residues below the orthosteric binding pocket form a new packing stack to stabilize the active conformation of lysophospholipid receptors.

Analysis of the single-particle cryo-EM data enabled us to capture distinct modes of interaction between the receptor and the G-protein. For both LPA₁ and S1P₁, the ICL2 region is an unstructured loop that is uncommon among Class A GPCRs (Supplementary Fig. 10). Despite that the interactions between ICL2 of the receptor and the αN of Gi is less extensive in these lysophospholipid receptors, the side chain of a leucine residue is inserted into the conserved hydrophobic pocket at the end of αN (Fig. 7). Additionally, in the S1P₁–Gi complex, salt bridges and hydrogen bond networks from ICL2 and ICL3 of the receptor to the α5 helix of Gi, contributing further to the stability of the complex (Fig. 7). Furthermore, we discovered that ICL2 of LPA₁ samples interactions with both the αN and α5 helices of the G-protein, while the C-terminal residue F354 of Gαᵢ is capable of interacting with either TM6 or ICL4 of the receptor (Fig. 8). Future studies will reveal the effect of these interaction modes in the process of G-protein activation by GPCRs.

The first FDA-approved lysophospholipid receptor modulator was Fingolimod (also named Gilenya or FTY720), an immuno-modulator approved in 2010 as the first oral treatment of MS[7,39–41]. Effectiveness of Fingolimod spurred the generation of other S1P$_1$ subtype-specific modulators for the treatment of MS, inflammatory bowel disease, psoriasis, and systemic lupus erythematosus[25,41–44]. S1P$_1$ is essential for lymphocyte egress from lymphoid organs into systemic circulation and provides a well-defined drug target for autoimmune disorders[39,42]. The binding of MS drugs to S1P$_1$ is the one that contributes to the mechanism of action[39]. The most serious adverse effects of Fingolimod are bradycardia and atrioventricular block, which are caused by the non-selective binding of Fingolimod to other subtypes of S1P receptors, particularly S1P$_3$[44]. Therefore, the goal for future drug development is to improve drugs that can bind more selectively to S1P$_1$. Given the high amino acid sequence homologies (Fig. 6), we modeled the structures of other S1P receptors based on the S1P$_1$ structure described here. From the orthosteric ligand binding pockets of these receptors, our data provides a clear structural explanation why Siponimod selectively binds to S1P$_1$ and S1P$_5$, but not S1P$_2$, S1P$_3$ and S1P$_4$ (Fig. 6). Residues F274 in S1P2, F263 in S1P3 and M289 in S1P$_4$ would clash with Siponimod binding (Fig. 6). As shown in Figs. 2 and 6, the B3 and B2 sites (different between S1P$_1$ and S1P$_5$) could be explored to improve the selectivity for S1P$_1$. Hence, our structural information will facilitate the development of next-generation S1P$_1$ selective therapeutics.

## Methods

**Expression and purification of LPA$_1$, S1P$_1$, Gα$_i$, Gβ$_1$ and Gγ$_2$.** A HA signal peptide and a Flag tag were fused to the human LPA$_1$ (2-340) or human S1P$_1$ (2-347), followed by the PreScission protease cleavage site, eGFP and 8xHis tag at the C-terminus. The construct was expressed and purified from *Spodoptera fru-giperda* Sf9 insect cells grown in ESF 921 protein-free medium (Expression Systems). After infections for two days, cells grown to approximately 2-3 million per ml were harvested and frozen at -80 °C until use. Thawed cell pellets were lysed with 2% n-Dodecyl-β-D-Maltopyranoside (DDM, Anatrace) and 0.4% Cholesteryl Hemisuccinate (CHS, Anatrace) in TBS buffer containing 20 mM Tris (pH 8.0), 150 mM NaCl supplemented with protease inhibitors (2 mg/ml leupeptin, 0.8 mM aprotinin and 2 mM Pepstatin A, Goldbio) and 0.1 mM EDTA at 4 °C. Insoluble particles were removed by ultracentrifugation at 142,000 g for 1 hour at 4 °C. The supernatant was then incubated with house-made GFP nanobody beads, washed wish TBS in 0.05% DDM and 0.005% CHS, and cut by PreScission protease over night at 4 °C. Eluted LPA$_1$ was further purified through a Superdex 200 Increase 10/300 column (GE Healthcare) equilibrated with TBS in 0.05% MNG, and peak fractions were pooled and concentrated for complex assembly.

N-terminally His-tagged full-length rat Gα$_i$ dominant negative (G203A) proteins were expressed and purified from *E. coli* strain BL21 (DE3)[45]. Cells were grown in 2xYT medium at 37 °C and induced by 0.2 mM IPTG for 16 hours at 20 °C before harvested. Wild type bovine Gβ$_1$ and C-terminally His-tagged soluble Gγ2 (C68S) proteins were co-expressed and purified from Sf9 insect cells[46]. Cells grown to 2-3 million per ml after infection were harvested. Gα$_i$ and Gβ$_1$γ$_2$ were purified virtually using the same procedure except that Gα$_i$ was constantly in buffers with 10 μM GDP (MilliporeSigma). Cells were lysed by sonication in buffer supplemented with 2 mM MgCl$_2$, 2 mM β-mercaptoethanol (β-ME) and 1 mM phenylmethylsulfonyl fluoride (PMSF). After centrifugation, the supernatant was purified through Ni-NTA (Qiagen) column. The eluate was loaded to a Superdex 200 Increase 10/300 column to pool the peak fractions, which was then concentrated for complex assembly.

**LPA-LPA$_1$-G$_i$, S1P-S1P$_1$-G$_i$ and Siponimod–S1P1–Gi complex assembly and purification.** Purified LPA$_1$ or S1P$_1$, Gα$_i$ and Gβ$_1$γ$_2$ were mixed at a ratio of 1:1.5:1.5 in HBS containing 10 mM HEPES (pH 7.0), 100 mM NaCl, 0.05% MNG, 0.01% glycerol and 0.1 mM TCEP, and incubated with 0.4 U Apyrase (Sigma-Aldrich) and 2 mM MgCl$_2$, in the presence of 10 μM LPA (Avanti), or S1P/Sipo-nimod (Cayman). After incubation at 4 °C overnight, the mixture was purified again through a Superdex 200 Increase 10/300 column in HBS with 1 μM desired agonist. The assembled complex was pooled and concentrated to 1.5 mg/ml for making cryo-grids.

**Cryo-EM data collection.** The LPA$_1$–G$_i$ or S1P$_1$–Gi complex was applied to glow-discharged 400 mesh gold Quantifoil R1.2/1.3 holey carbon grids (Quantifoil Micro Tools) and vitrified by Vitrobot Mark IV (Thermo Fisher Scientific/FEI).

Micrographs were collected with SerialEM using beam-image shift of 5×5 on a 300 kV Titan Krios electron microscope (Thermo Fisher Scientific/FEI) at a nominal 22,500× magnification with a Gatan K3 direct electron detector (Gatan, Inc.)[47,48]. In total, 10,500 movies in the defocus range of −0.8 to −2.2 μm were recorded with a total accumulated dose of 28.2 e$^-$/Å$^2$ using SerialEM 3.7 and Leginon 3.5.

**Image processing, 3D reconstructions, modeling, and refinement.** Please refer to the workflow diagrams for exact software versions and details used in processing. Movie stacks were motion-corrected by MotionCorr2 v1.2.1, and ctf was estimated using CTFfind v4.1.10. For each dataset, Relion 3.0 Laplacian-of-Gaussian picking with minimum and maximum dimensions of 76 Å and 119 Å was used to heavily over-pick at a rate of approximately 2000 particles per micrograph. The resulting particle stacks of 4-17 million particles were fourier-cropped and processed through multiple rounds of heterogeneous classification in CryoSparc v2.14.2 and v2.15.0[49], steadily decreasing the cropping factor as false positives were removed and resolution improved (Supplementary Figs. 1–3). 2D classification confirmed that the majority of particles excluded by heterogeneous classification were false positives, receptor alone or G-proteins alone. The final stacks of intact complexes contained between 0.4-2.2 million particles. The consensus stacks were subjected to Local CTF Refinement procedures in CryoSparc followed by Bayesian Polishing in Relion[50,51], and finally Global CTF Refinement in CryoSparc (Supplementary Figs. 1–3). The consensus stacks were subjected to Local Refinement in CryoSparc for the GPCR and heterotrimer independently. The Local Refinement maps showed significantly improved features over the consensus maps, all with resolutions below 2.6 Å (Supplementary Figs. 1–3). The consensus and two local refinements for each dataset were finally density-modified in Phenix v1.17.1-3660 without a reference to correct errors in Fourier coefficients[52]. The density of the α-helical domain of Gα$_i$ was poor. A model was built starting from 4Z34 and 1GP2, and was used to generate a composite map in Phenix from the density-modified consensus and local refinements. This model was real-space refined against the composite map, and a work/free half-map pair was used to ensure against over-fitting.

3D variability analysis (3DVA) was performed for each consensus particle stack to investigate the main components of heterogeneity present (Supplementary Movies)[33]. For LPA$_1$–G$_i$, 3DVA clustering was used to generate 20 maps. Initial models were generated by rigid-body fitting each chain from the consensus model into the density. Visual inspection of the models revealed two extreme positions for the N-terminal helix of G$_i$ relative to LPA$_1$. Selecting the clusters corresponding to the most extreme position, Non-Uniform Refinement as well as local refinements about the GPCR and G-proteins were used to generate density-modified composite maps, as described above (Supplementary Figs. 5 and 11)[53]. The consensus model was used as a starting point to build the new models and refine them against the two new maps generated for State a and State a'. Work/free half-map pairs were used to ensure against over-fitting. All models were created in Coot v0.8.92 and refined against maps in Phenix v1.17.1-3660. Figures were generated using Chimera v1.14 and Pymol 2.5.

**cAMP measurement.** For LPA$_1$, B103 cells stably expressing wild-type or mutant human LPA$_1$ were seeded on collagen I coated 24-well plates (Gibco) and serum starved overnight. For S1P$_1$, CHO cells were transiently transfected with wild-type or mutant S1P$_1$. The cells were washed twice with Hank's balanced salt solution containing 25 mM HEPES-NaOH (pH 7.4) and 0.1% BSA and incubated in the buffer containing 0.5 mM IMBX (Sigma) for 20 min at room temperature. Cells were then stimulated with 10 μM forskolin for 30 min at room temperature, in the absence or presence of various concentrations of agonist (Cayman). The reaction was terminated by aspiration of medium and immediately treated with 0.1 M HCl for 10 min at room temperature. Cells were harvested by centrifugation, and the supernatant was collected for the determination of cAMP concentration in tripli-cate with the Direct Cyclic AMP Enzyme Immunoassay kit (Enzo Life Sciences). The activity was expressed as the percentage of forskolin-induced cAMP accu-mulation. Membrane receptor expressions in these stably transfected cells were measured by Western blots and were found to be at similar levels. The cAMP assays were repeated three times, and the data are represented as mean ± SEM of the three independent experiments. The analysis was done using the log(agonist) vs. response function of Prism 8 (GraphPad)[46,54].

**LPA ligand-binding assay.** B103 cells stably expressing Wild-type or mutant human LPA$_1$ plasmids were transfected into B103 cells and stable cell lines were established. Cell surface expression of LPA$_1$ was measured by flow cytometry: WT, 71%; K39A, 99%; T113A, 95%; R124A, 97%; Q125A, 52%; D129A, 34%; K294A, 93%; vector control, 3%. A free solution assay, where the receptor (LPA$_1$/mutants) containing nanovesicles (of 110-130 nm size as measured by DLS) and unlabeled 18:1 LPA ligand are freely moving into solution, was used in a native environment of the binding partners (18:1 LPA-LPA$_1$). The assay was conducted with Com-pensated Interferometric Reader (CIR) that measured light refractive index change from binding-induced conformational and/or hydration changes produced by real time binding events in a *sample* (receptor containing nanovesicles + 18:1 LPA) compared to a non-binding *reference* (RI matched buffer + 18:1 LPA). The

interferometric signal from vector nanovesicles binding to 18:1 LPA (non-specific) was subtracted from the LPA$_1$ or its mutants-containing nanovesicles binding to 18:1 LPA (total). 18:1 LPA (Avanti Polar Lipids) dilution series were prepared in a 0.01% fatty-acid free BSA/0.002% EtOH/PBS (pH7.4) solution. Mixture of EtOH/BSA was used to maximize LPA solubility in solution, since LPAs have poor solubility and tendency to adhere in the Eppendorf wall when prepared and successively diluted using only aqueous buffer. Total protein concentration was maintained at 25 ug/ml. The Compensated Interferometric Reader is a benchtop refractive index (RI) reader that combined a compensated interferometer (CI) with a Mitos Dropix (an automated droplet generator) and a syringe pump. The compensated interferometer which consisted of a diode laser, one or two mirrors, one glass capillary and a CCD camera, measures RI change between the binding *sample* and *reference* from positional shift in backscattered interference fringes produced from the interaction between an expanded beam profile of the laser and a capillary filled with droplets of *sample-reference* solutions. The positional shift of the backscattered fringes, which is equivalent to molecular interaction, was quantified using FFT (fast Fourier transform) of selected fringes in a CCD camera. The LPA concentration dependent change in RI ($\Delta$RI) signal to LPA$_1$/vector or LPA$_1$ mutants was fitted using the single site total vs non-specific binding isotherm using GraphPad Prism.

**Quantification and statistical analysis**. The cAMP assays were repeated three times, and the data are represented as mean ± SD of the three independent experiments. Cryo-EM data collection and refinement statistics are listed in Supplementary Tables 1 and 2.

**Reporting summary**. Further information on research design is available in the Nature Research Reporting Summary linked to this article.

## Data availability

Source data are provided with this paper. The cryo-EM density maps and corresponding coordinates have been deposited in the Electron Microscopy Data Bank (EMDB) and the PDB, respectively, under the accession codes: EMD-25819 (LPA–LPA$_1$–Gi), EMD-25820 (LPA–LPA$_1$–Gi state a), EMD-25821 (LPA–LPA$_1$–Gi state a'), EMD-25822 (S1P–S1P$_1$–Gi), EMD-25823 (Siponimod–S1P$_1$–Gi), and PDB 7TD0 (LPA–LPA$_1$–Gi), 7TD1 (LPA–LPA$_1$–Gi state a), 7TD2 (LPA–LPA$_1$–Gi state a'), 7TD3 (S1P–S1P$_1$–Gi), 7TD4 (Siponimod–S1P$_1$–Gi). Source data are provided with this paper.

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

## Acknowledgements

We thank M. de la Cruz at The MSKCC Richard Rifkind Center, the MSKCC HPC group for assistance with data processing and members of our research groups for helpful discussion and comments on the manuscript. This work was supported by NIH grants GM138676 (X.Y.H.), CA243235 (N.P.), GM132307 (R.K.H), NIH-NCI Cancer Center Support Grant P30 CA008748 (R.K.H), the Josie Robertson Investigators Program (R.K.H.), the Searle Scholars Program (R.K.H.), NIH NS103940 (Y.K.), NS084398 (J.C.), and DoD W81XWH-17-1-0455 (J.C.).

## Author contributions

S.L. expressed and purified LPA$_1$, S1P$_1$, Gα$_i$, Gβ$_1$γ$_2$, and the protein complexes, made cryo-EM grids, performed cryo-EM screening, data collection, model building, and manuscript preparation. N.P. made cryo-EM grids, performed cryo-EM screening, data collection, EM density map determination, analysis of dynamics, and model building under the supervision of R.K.H. L.Z. performed cAMP assays under the supervision of W.L. Y.K. and M.R. established the B103 cell lines stably expressing the LPA$_1$ mutants and performed the ligand-binding assays under the supervision of J.C. X.Y.H. supervised the project, interpreted data, and wrote the manuscript. All authors contributed towards the final version of the manuscript.

## Competing interests

The authors declare no competing interests.
