## [Peer Review File · Nature Communications]

Differential Activation Mechanisms of Lipid GPCRs by Lysophosphatidic Acid and Sphingosine 1-PhosphateEditorial Note: This manuscript has been previously reviewed at another journal that is not operating a transparent peer review scheme. This document only contains reviewer comments and rebuttal letters for versions considered at *Nature Communications*.

REVIEWERS' COMMENTS

Reviewer #1 (Remarks to the Author):

Original review for [another journal (redacted)]

Single particle cryo-EM imaging is used to analyze GPCR signaling complexes comprising human S1P1 and Gi in the presence of S1P, human S1P1 and Gi in the presence of the MS drug Siponimod, and human LPA1 and Gi complexes formed with bound LPA. Using RELION and cryoSPARC, the complexes are analyzed using a collection of tools, namely, consensus refinement, local refinement (of GPCR receptor and the G-protein subunits), 3D variability to assess conformational heterogeneity due to interaction between receptors and the G-proteins, and nonuniform refinement of specific conformational states of the G-proteins.

The introduction motivates and work well. The results are described clearly and in detail, and the methodology used, including the image analysis, is well described. As such, I very much liked the paper, and felt the work was significant.

One aspect of the work about which I would have liked to have seen more detail concerns the use of 3D Variability Analysis to explore differences between the interaction dynamics of S1P1-Gi and LPA1-Gi. In particular, for the analysis of LPA1-Gi, 3D Variability Analysis was used to visualize the conformational variability in the interaction of the receptor and the G-protein, and to isolate two 'extreme' conformational states of the G-protein. The paper describes the motion of the G-protein (along with movies), as well as the changes in conformational states. Nevertheless, it might also be interesting to see the distribution of latent coordinates from 3D Variability Analysis, which can be interpreted as a representation of the conformational landscape of the particle. Also, in the latent coordinate domain, it would have been interesting to see the 20 conformations found via clustering, to what extent these clusters are well separated, and where the two extreme conformations, a and a', are in this space.

While the results of 3D Variability Analysis are described in some length for LPA1-Gi with bound LPA, the conformational changes of which are nicely visualized in the movies provided, a similar analysis of S1P1-Gi, either with bound S1P or bound Siponimod, is not provided. Nor are movies showing conformational variation for the S1P1-Gi complexes. It might be of interest to show movies from 3D Variability Analysis

for these other complexes along with the distribution of their latent coordinates (the conformational landscapes). This would allow direct comparison with LPA1-Gi with bound LPA.

The authors' response to the original reviews, provided with this re-submission to Nature Communications has very nicely addressed any concerns or questions I had. I think this work is a valuable contribution, and recommend that the paper be published.

Reviewer #2 (Remarks to the Author):

Authors have satisfactorily addressed the points raised on the original manuscript, and the revised manuscript should be published in Nature Communications.

Reviewer #3 (Remarks to the Author):

The authors have attempted to address previous reviews by adding an additional discussion and a new figure, which provides a further comparison of their structural data. Even though the work does confirm prior published work, the structural insights from this study are not confirmed by experimental studies, and therefore are speculative in nature. I am not certain what additional insights are learned from the authors' work on LPAR vs. S1PR - other than what is known already. Their work does contain structures of receptors complexed with heterotrimeric G proteins which do provide additional information. However, novel new insights from this work are limited at this stage of development.

List of Manuscript Changes

We thank the reviewers very much for the helpful comments on our manuscript.

Reviewer #1:

“The authors' response to the original reviews, provided with this re-submission to Nature Communications has very nicely addressed any concerns or questions I had. I think this work is a valuable contribution, and recommend that the paper be published.”

Thank you!

Reviewer #2:

“Authors have satisfactorily addressed the points raised on the original manuscript, and the revised manuscript should be published in Nature Communications.”

Thank you!

Reviewer #3:

“The authors have attempted to address previous reviews by adding an additional discussion and a new figure, which provides a further comparison of their structural data. Even though the work does confirm prior published work, the structural insights from this study are not confirmed by experimental studies, and therefore are speculative in nature. I am not certain what additional insights are learned from the authors' work on LPAR vs. S1PR - other than what is known already. Their work does contain structures of receptors complexed with heterotrimeric G proteins which do provide additional information. However, novel new insights from this work are limited at this stage of development.”

Although the S1P₁/Gi structures were scooped by another group while our manuscript was under review, our structural comparisons of the LPA₁/Gi and S1P₁/Gi complexes go beyond a structural study on one type of lipid GPCR. We have revealed that, even though LPA and S1P have similar chemical structures, they adopt different conformations to activate their receptors. Furthermore, even though LPA₁ and S1P₁ are similar lipid GPCRs, they are activated differently by the lipid ligands. These are all new insights from our work.